# Ten Years of CRISPRing Cancers In Vitro

**DOI:** 10.3390/cancers14235746

**Published:** 2022-11-23

**Authors:** Davide Capoferri, Serena Filiberti, Jessica Faletti, Camilla Tavani, Roberto Ronca

**Affiliations:** Section of Experimental Oncology and Immunology, Department of Molecular and Translational Medicine, University of Brescia, 25121 Brescia, Italy

**Keywords:** cancer, hallmark, CRISPR, in vitro, genome editing, technology

## Abstract

**Simple Summary:**

There are several ways to mimic cancer cells features, one of those being permanently editing their DNA. Even though cancer cells alone cannot represent the whole complexity that develops around them in their surroundings, their modification, characterization and employment in rather simplified tests constitutes a fundamental step prior to contextualize them in living models, such as mice, both to comply with the 3Rs rule, and to optimize the in vivo works. On such notes, this review aims to highlight all the processes and discoveries with a long-term intention to make cancers more curable.

**Abstract:**

Cell lines have always constituted a good investigation tool for cancer research, allowing scientists to understand the basic mechanisms underlying the complex network of phenomena peculiar to the transforming path from a healthy to cancerous cell. The introduction of CRISPR in everyday laboratory activity and its relative affordability greatly expanded the bench lab weaponry in the daily attempt to better understand tumor biology with the final aim to mitigate cancer’s impact in our lives. In this review, we aim to report how this genome editing technique affected in the in vitro modeling of different aspects of tumor biology, its several declinations, and analyze the advantages and drawbacks of each of them.

## 1. Introduction

Clustered Regularly Interspaced Short Palindromic Repeats (CRISPR) are short sequences in prokaryotic genomes discovered long before the omics era [1], the role of which has been explored and defined later in time [2] as an adaptive immunity strategy in bacteria. Indeed, analogously to eukaryotic adaptive immunity, it involves a genome rearrangement whose consequence is the triggering of a pathogen-specific response that impairs the biology of the pathogen itself, in an attempt to eliminate the causes of the immune response. This parallelism has been clearly explained by Wiedenheft et al. [3].

In the prokaryotic world, the main environmental pathogens that act against bacteria are those viruses called bacteriophages or phages. The strategy of the phages involves the attachment to the bacterial cell wall and the injection of viral DNA: viruses are not able to self-replicate; thus, the aim of this attack is to hack the bacterial machinery for DNA replication and protein translation and generate new viral particles. Bacteria, though, are not defenseless: they have at their service the CRISPR-associated (Cas) gene enzymes system, which includes polymerases, nucleases and helicases. Cas proteins can detect exogenous double stranded DNA (dsDNA), break it into small pieces (usually around 30 bp long) and integrate it in the CRISPR array sequence. The transcription of these sequences is then processed as short CRISPR RNAs (crRNAs), which are attached to a transactivating crRNA (tracrRNA) forming a single guide RNA (sgRNA), loaded onto other Cas protein complexes that, through sequence homology, can target and specifically cleave foreign dsDNA sequences upon re-exposure [4], generating a double strand break and then disabling the phagic threat.

With a close look on cancer, the method taken by molecular biology to engineer CRISPR-associated proteins and its history has been elegantly described by Zhang et al. [5], the use of different strategies by Moses et al. [6] while the first in vivo applications of CRISPR genome editing by Kannan and Ventura [7]. The purpose of this review is to evaluate, 10 years after its first description as a tunable strategy, the impact that CRISPR technology has had so far on cancer research.

## 2. CRISPR and Cancer

Cancer is a generic term that encompasses more than 270 different diseases. Globally it is one of the top deadliest diseases, ranking as the first or second cause of premature death in 134 of 183 countries and third or fourth in another 45 [8]. Virtually any type of human cell can transform and become a tumor cell, but despite this huge variability some features are commonly found in all these neoplastic diseases and are referred to as “the hallmarks of cancer”, as reported in the pioneering reviews of Hanahan and Weinberg [9,10,11]. Knowledge of these characteristics is fundamental when seeking for actionable targets and experimental readouts, especially within that range of neoplasms that are difficult to tackle regardless of the efforts made by research groups and clinicians so far. In the next sections, the most significant research output obtained by CRISPR application will be described accordingly to categories that summarize diverse hallmarks. To facilitate the reader, we generated Figure 1 to associate each hallmark with the considered genes and the effect that CRISPR-mediated genome editing had within that model.

### 2.1. Life and Death of Tumor Cells

Within this category we grouped those hallmarks that directly manage cell growth, proliferation, apoptosis and senescence.

#### 2.1.1. Sustain Mitogenic Signaling

The ability to proliferate “without control” is maybe the most widespread known feature of cancer cells. The mechanisms through which uncontrolled proliferation occurs have been widely investigated prior to the advent of CRISPR, though this technique allowed to dig further into proliferation phenomena. Table 1 gathers the best characterized genes and their proliferation-associated mechanisms interrogated by CRISPR during the last ten years.

Regarding what concerns extracellular growth-promoting signals, a particular focus should be put on growth factors and their receptors.

Chen et al. investigated the impact of AZD4547, a FGFR inhibitor, on FGFR2-amplified gastric cancer cell line KatoIII, using a kinome-wide CRISPR knock-out screening panel. Comparing FGFRi-resistant cells to untreated cells, ILK (integrin-like kinase) was classified as the most significantly implicated in AZD4547 resistance, and downstream signaling of this kinase was investigated. Phosphorylation of GSK3β was identified as a potential target downstream of ILK whose inhibition in cooperation with AZD4547 might increase the effectiveness of FGFR inhibitors against FGFR2-amplified gastric cancers [32].

The role of EGFR in renal cancer was the object of study of Liu et al., who used RC21 cells as disease model. Knock-out of EGFR highlighted its role as promoter of proliferation, as its absence resulted in the arrest of cell cycle at G2/M checkpoint. However, the KO cells showed a decreased sensitivity to cisplatin and to the HDAC inhibitor SAHA. Further, knock-outs had a basally higher ERK1/2 phosphorylation status, which could be decreased upon treatment with TKi sutininib or growth factor PDGF. This study concluded that EGFR inhibition in combination with sutininib might have a better outcome for EGFR-expressing renal cell carcinoma patients [33].

Lee et al. instead focused on the ligand HGF, and generated knock-outs of two hepatocellular carcinoma cell lines, Huh7 and Hep3B. Both KO cell lines showed an overall decrease in viability, motility and clonogenicity, together with a reversion to a more “epithelial” phenotype, as shown by the reduction in N-cadherin and vimentin and the increase in E-cadherin levels. Activated second messengers, such as phosphorylated Met, p38, Akt, ERK and Jnk, were reduced upon KO of HGF, and upon exposure of WT HCC cell lines to H_2_O_2_ showed a greater resistance to apoptosis, highlighting the role of HGF autocrine signalling not just on cell proliferation [34].

Moreover, the CRISPR/Cas9 knock-in approach has been used in glioblastoma cell lines to analyze the cellular proliferative rate by the insertion of a proliferation reporter, that enabling the monitoring of cell division permitted the study of tumor cell quiescence [35].

#### 2.1.2. Resist Growth Suppression

When we think of a tumor cell, the progression towards malignant phenotypes does not limit resistance to apoptotic signals or the enhancement of mitosis, but it can involve the elusion of those signals that prevent the cellular system to proliferate. Usually, these signals are made of antimitogenic stimuli, soluble or anchored onto cell membranes, and responses to genomic damages. All of these signals exert their effect by blocking cell cycle progression. In this section, we will focus on the antimitogenic stimuli (transforming growth factor beta, TGFβ) and on cell cycle checkpoints represented by Retinoblastoma proteins (Rbs), p53, BRCA1/2 and LKB1.

Antimitogenic signals from TGFβ do not affect only TGFβ-sensitive tumor cells, but represent a widespread signal present in the microenvironment that can also affect immune cells, preventing them from proliferating. The only reported case of CRISPR employment against TGFβ pathway for cancer-related purposes dates to 2020 and was used to knock-out TGF-BRII receptor in Chimeric Antigen Receptor T cells (CAR-T cells), making them insensitive to microenvironment TGFβ and allowing for a longer-lasting solid tumor-cytotoxic response [36].

Oncosuppressor protein Rb1 exerts its role by binding to transcription factor E2F when the former is not phosphorylated. Phosphorylation of Rb1 allows the release of E2F, which in turns binds to promoters and initiates the transcription of those genes required for proceeding to phase S [37].

Retinoblastoma models are difficult to establish, therefore Kanber et al. inactivated *RB1* by CRISPR in H9 human embryonic stem cells, which then were differentiated to retina cells. *RB1*-KO differentiating cells showed already an impaired development of the retinal phenotype, with a persistent proliferation rate, failure to generate organoids and their disintegration 126 days after the beginning of differentiation. RNAseq highlighted the upregulation of proliferation-associated genes as well as retinoblastoma-related oncogenes, *DEK*, *SYK* and *HELLS* above others [38].

Marshall et al. instead generated single and double copy KO of Rb1 in sarcoma U2OS line, NSCLC cell line H460 and lung adenocarcinoma cell line H1792. Those cell lines showed an increased sensitivity to oxidative and genomic damage, confirming that Rb1-deficient cancers would benefit from platinum-based therapies; they also showed homologous recombination repair defects and an increase in lung metastases in vivo [39].

Further, Oser et al. used CRISPR screening on the challenging small cell lung cancer modeled by the cell lines H82 and H69. This led to the identification of the hyperdependence of Rb1 from Aurora Kinase B, which is druggable, and the molecule used during this investigation for AURKB showed a low toxicity [40].

Chakraborty et al. generated prostate cancer LNCaP cells with a double knockout of *BRCA2* and *RB1*, following the rationale by which in aggressive prostate cancer BRCA2 is often co-mutated together with Rb1. Deletion of *BRCA2* alone increased the resistance to PARPi drugs, while the double KO induced an invasive phenotype and EMT markers upregulation [41].

Tang et al. investigated the impact of *mutant TP53* knockout in osteosarcoma cell lines KHOS and KHOSR2. In these models, the lack of *mutant* p53 resulted in less proliferative cells, lower levels of the antiapoptotic proteins Bcl-2 and survivin, the reduction of the oncogene *IGF-1R*, lower motility and an increased sensitivity to the DNA-binding chemotherapeutic doxorubicin [42].

Although those two proteins had been already identified prior to the advent of CRISPR, their gene editing demonstrated that the molecular mechanism they govern might represent valuable druggable targets.

LKB1 is an oncosuppressor gene, whose mutations occur in one in every five NSCLC patients and whose activity mostly depends on the phosphorylation of AMP-dependent protein kinases. Hollstein et al. dedicated a full in vitro to in vivo work to identify and demonstrate that salt-inducible kinases SIK1 and SIK3 are the two most relevant LKB1-regulated kinases that redundantly mediate its oncosuppressor activity in KRAS-driven lung cancer cells. Upon generating by CRISPR a KO per each of the 14 kinases known to be activated by LKB1, plus some combinations including the restore of LKB1 expression in NSCLC LKB1-null A549 cells, *SIK1* alone and *SIK1*/*SIK3* knock-outs resulted with the ability to recapitulate almost in full the LKB1-null features, and the complementary in vivo study confirmed and further characterized the impact that those two kinases have on LKB1-dependent signaling [43].

#### 2.1.3. Limitless Replicative Potential

Limitless replicative potential is one of the most well-established hallmarks of cancer, being part of the original six acquired capabilities of tumor cells proposed in 2000 [9]. This trait is essential to ensure expansive tumor growth and dissemination. Although three of the acquired capabilities described above—self-sufficiency in growth signals, insensitivity to anti-growth signals, and resistance to apoptosis—lead to a deregulated proliferation program, this is not sufficient to enable the generation of the vast cell population that constitute macroscopic tumors. Indeed, healthy cells present a limited replicative potential because they have a definite number of allowed doublings. Therefore, during the multistep cancer progression, premalignant cell populations overtake the limit of allowed doubling by achieving an immortal status [44].

Cell immortalization is gained through reactivation of telomere maintenance mechanisms that consists of two different pathways: the first one is telomerase-dependent, the second is the telomerase-independent alternative lengthening of telomeres (ALT) [45].

Telomerase is a ribonucleoprotein made of two different subunits, a reverse transcriptase (TERT) in complex with a long ncRNA called telomerase RNA component (TERC), that contains the template region for telomere synthesis. TERC is constitutively transcribed in somatic cells, while TERT is normally suppressed. In total, 80–90% of cancer cells activate TERT and subsequently telomerase to reach immortality [46].

CRISPR-Cas9 technology has been employed for gene editing therapy by targeting telomerase in cancer cells. Wen et al. employed it to create TERT-KO HeLa, PANC-1 and SUM159 cell lines. Genotyped HeLa clones were tested in vitro and in vivo, and showed an impairment in growth, a more apoptotic phenotype and a greatly reduced ability to develop tumor masses when injected intramuscularly in nude mice. To build the model, the authors developed a dual gRNA gene knockout strategy in order to overcome two potential risks: the odds of getting in-frame indels that may not necessarily lead to gene knockout, and the chance that the induced indels would introduce cancer-promoting mutations. This strategy aimed at targeting the introns surrounding TERT exon 4: and by doing this, exon 4 was entirely removed [47].

Facing the cancer telomerase issue from a different perspective, a couple of groups used Cas9 to edit the promoter sequences of TERT, providing a useful tool for studying telomerase biology and understanding the molecular basis by which cancer-associated TERT mutations impact telomerase activity [48,49].

These groups have encountered some problems editing TERT sequences, because it is a locus with low targeting efficiency. In fact, TERT is not a very actively transcribed gene and its chromatin conformation prevents the access of the Cas9–sgRNA complex. Moreover, the high GC content around the TERT 5′ region blocks target recognition by the Cas9–sgRNA complex. To overcome these limits, researchers developed a “pop-in/pop-out” approach; this strategy consists in the introduction of a single base-pair substitution into the TERT promoter alongside an eGFP expression cassette, which is then removed by a second round of CRISPR-mediated editing, resulting in a TERT promoter with only a single base-pair alteration. Thanks to this strategy, by reverting C124T mutation in SCaBER cells, a model of urothelial cancer, Xi et al. managed to decrease telomerase levels, therefore obtaining telomere shortening and a reduced proliferation rate [49].

Telomerase represents an excellent target for cancer therapy. Different telomerase activity inhibitors have been developed to treat cancer, but all failed due to side effects. Thanks to CRISPR-Cas9 technology, a cancer gene therapy named telomerase-activating gene expression (Tage) system was developed to target telomerase in cancer cells. The Tage system consists of multiple sequences that are subsequently activated only in those cells with an active telomere synthesis: it all gets triggered by the pairing of the endogenous telomerase to a single stranded sequence upstream of a wild type Cas9, which makes this enzyme elongate this sequence by adding telomeres. Moreover, a single guide RNA that recognizes telomeres and a dead Cas9 (without any nuclease ability) tagged with VP64 are produced under different promoters. The abundance of telomere-targeting sgRNA and dCas9-VP64 allows their pairing, and the complex binds to the telomeres of the formerly described construct. VP64 is a transcriptional activator, and the serial deposition of dCas9-VP64 chimeres upstream of a wild type Cas9 greatly enhances its transcription. As a result, the abundance of wild type Cas9 competes with dCas9-VP64 for the telomere-targeting sgRNA, and this huge amount can sustain the targeting of nucleolar telomeres, cleaving them and killing the cell. Upon some rounds of optimization, the authors effectively induced death in various cancer cells tested (HepG2, HeLa, PANC-1, MDA-MB-453, A549, HT-29, SKOV-3, Hepa1-6) without affecting normal cells (MRC-5, HL7702) and bone marrow mesenchymal stem cell [50].

While most tumors reactivate telomerase expression to become immortal, the expression of TERT is finely regulated and rate-limiting for telomerase activity maintenance. Several general transcription factors have been found in regulating TERT transcription; however, a systematic study of positive regulators of TERT expression was lacking. Thanks to an inducible CRISPR/Cas9 KO screen, using two sets of sgRNA libraries, respectively targeting nuclear genes and genes of unknown function it had been possible to reveal multiple candidates that could upregulate TERT expression. The most suitable is a E3 ubiquitin ligase DTX2. The generation of clones that are *DTX2* KO in telomerase-positive cancer cells significantly reduced TERT mRNA levels as well as telomerase activity, resulting in progressive telomere shortening, cell growth arrest and increased apoptosis [51].

About 10% of cancers lack telomerase activity and adopt a different strategy to maintain telomere lengths: namely the telomerase-independent Alternative Lengthening of Telomeres (ALT) pathway and is a telomere-specific mechanism of homology directed repair [52]. Graham et al. used the CRISPR-Cas9 system to generate gene KO of α-thalassemia/mental retardation X-linked protein (*ATRX*) in two prostate cancer cell lines, one telomerase-positive and the other telomerase-negative. *ATRX* mutation is strongly associated to ALT-positive cancers, as it can contribute to the recruitment of H3.3 histone which maintains the heterochromatic morphology of the genome at telomeres level. *ATRX* KO induces ALT activation in the telomerase-negative cancer cells but, alone, is not sufficient to activate ALT in telomerase-positive cancer cells, in which it was necessary to also create gene KO of TERT through the CRISPR-Cas9 system to achieve comparable results [53].

#### 2.1.4. Resistance to Apoptosis

Evading apoptosis is one of the leading behaviors adopted by cancer cells in order to promote their growth and development. Almost all cells are able to activate the apoptotic program according to intra- and extra-cellular signals and this is a fundamental process that guarantees tissue maintenance. At the same time, resisting cell death is an essential capability for cancer cell survival [9]. Some principal strategies for resisting apoptosis are loss of *TP53* suppression function, increased expression of antiapoptotic regulators (BCL-2, BCL-XL) and proapoptotic factors downregulation (BAX, BAK, BIM, PUMA, NOXA) [10].

CRISPR has been employed to target genes involved in these pathways, both for a more thorough understanding of the signaling mechanisms underlying apoptotic-related phenomena and for the study of anticancer therapy resistance mechanisms. An example is CRISPR/Cas9-mediated deletion of hnRNP L binding site region in BCL2 3′UTR in a large-cell lymphoma cell line. hnRNP L is a factor that protects mRNAs from nonsense mediated mRNA decay, including BCL2 mRNA. hnRNP L binding site deletion reduces BCL2 expression and activity, and leads to apoptosis. Its knockout highlighted its role in avoiding apoptosis through BCL2 level modulation [54].

Genome editing has been used in a colorectal carcinoma cell line in order to investigate the molecular mechanism that links autophagy to apoptosis. CRISPR/Cas9 knock-out generation of critical autophagy regulators such as *ATG5*, *ATG7* and *FOXO3A* facilitated the understanding of how inhibiting autophagy can increase apoptosis sensitivity in response to antitumor drugs through the regulation of proapoptotic proteins such as PUMA and BIM [55].

Different studies examined the effects of the genetic inhibition of transcription factors on cancer cells. For example, CRISPR/Cas9-mediated deletion of NF-kB c-REL subunit in cervical carcinoma cells showed an effect on proliferative processes, but not on apoptotic ones [56]. On the other hand, tumor growth and progression are strictly connected to TNF signaling. CRISPR/Cas9 knockout of *IKK1/2*, a regulation factor complex in NF-kB signaling, leads to TNFα-induced cell death in HEK293 (human embryonic kidney cells) suggesting that IKK1/2 targeting could become a promising strategy in cancer therapy, in order to treat tumor cells resistant to TNFα-induced cell death [57].

In glioblastoma multiforme (GBM) cell lines, CRISPR/Cas9 genome editing technique led to identify different apoptosis-correlated genes: *ERN1*, *IGFBP3*, *IGFBP5*, *FAT1*, *CHAF1A*, *GLI1*, *TRIM45*, *RGS4*, *ATM*, *PDPN*, *ATG5*, *ATG7*, *C14-IP-3*. Their knock-out facilitated the understanding of their role in apoptotic mechanisms rather than as prognostic markers [58].

CRISPR/Cas9 knock-out of *ELOVL2*, a fatty acid elongase overexpressed in renal carcinoma cells responsible for cell proliferation, caused a significant activation of intrinsic apoptotic pathways, increasing the expression of BAX, BAK, PUMA and NOXA while downregulating the antiapoptotic factors BCL2 and MCL1 [59].

CRISPR/Cas9 loss-of-function screen identified that SKP2 and P27 expression is relevant for Bf3 cells resistance to CHK1 inhibitors. Inhibition of CHK1 indeed leads to BCL2-dependent death in tumor cells, therefore this CRISPR screening suggested further novel targets for increasing the therapeutic outcome of acute lymphoblastic lymphoma treatment [60].

Different CRISPR knock-out approaches (CRISPR-excision, CRISPR-HDR, CRISPR du-HITI) have been used in order to target LINC00511 lncRNA, a long intergenic non-coding RNA overexpressed in breast cancer cells. LINC00511 knock-out leads to important apoptosis rate increase, with an overexpression of proapoptotic genes such as *P57*, *P21*, *PRKCA*, *MDM4*, *MAP2K6*, *FADD* and downregulation of the antiapoptotic genes *BCL-2* and *BIRC5* (survivin) [61].

CRISPR/Cas9 knock-out screening has also been used to study the development of therapy resistance, gained through pro- and anti-apoptotic proteins disequilibrium. For example, in nasopharyngeal carcinoma (NPC), CRISPR/Cas9 screening associated to the use of BH3-mimetic drugs helped the identification of those crucial proteins that play an antiapoptotic role in NPC-related treatment resistance, namely BCL-XL, MCL-1 and BFL-1 [62].

Further, CRISPR knock-out screening facilitated the identification of BCL2L1 (*BCL-XL* gene) as a possible target in order to increase sensibility for gemcitabine in pancreatic cancer treatment [63], and *TP53* and *BAX* as key genes in venetoclax-resistance in acute myeloid leukemia cells [64].

Finally, CRISPR application and apoptotic pathways investigation have been employed also within the scope of chimeric antigen receptor (CAR) T-cell therapy improvement. Essential CAR-T resistance regulators have been identified by CRISPR screening and *NOXA* knock-out of lymphoma cell lines revealed a stronger CAR-T resistance and a lower apoptosis rate after CAR-T exposure [65].

#### 2.1.5. Cellular Senescence

Cellular senescence has long been seen as a fail-safe mechanism against oncogenic transformation, as expression of an oncogenic *RAS* gene in primary cells leads to a post-replicative state called oncogene-induced senescence. Moreover, even cancer cells can be induced to enter a state of senescence, not only as a result of chemotherapy treatment but also by excessive pro-survival and pro-proliferative signaling [11]. In contrast, a lot of evidence shows that in certain contexts, senescent cells stimulate tumor development and progression [66]. The principal mechanism by which senescence induces tumor phenotypes is due to the senescence-associated secretory phenotype (SASP), which consists of a complex mixture of extracellular vesicles, and bioactive proteins—including chemokines, cytokines and proteases—secreted by senescent cells [67]. The SASP represents a sort of double-edged sword with respect to tumor control. On one hand, the SASP can inhibit tumor growth by triggering an immunological response against it through the recruitment of the adaptive immune system. On the other hand, when senescent cells remain present in a tumor, the SASP can contribute to a chronic inflammatory response, which can result in acceleration of age-associated conditions and cancer metastases [68].

Beyond the debate on the role of cellular senescence with respect to tumor growth and progression, pro-senescence therapies are increasingly being considered for cancer treatment. Identifying new targets to induce senescence in cancer cells could further enable these therapies. In this context, CRISPR-Cas9 technology has been employed for the study and the screening of additional molecules and pathways that can be used as potential targets for senescence-inducing therapies.

Wang et al. used a loss-of-function CRISPR/Cas9-based genetic screen to identify new targets for inducing senescence in melanoma cells. Using a library of 5130 CRISPR vectors targeting 446 enzymes involved in chromatin remodeling and modulation of epigenetic marks, it was showed that loss-of-function of *SMARCB1* (a component of the SWI/SNF chromatin remodeler complex) induces senescence in A375, Mel888, Mel526 and Mel624 melanoma cells through strong activation of the MAP kinase pathway. As a matter of fact, cells harboring SMARCB1 loss-of-function are characterized by absence of proliferation markers, expression of tumor suppressor genes and presence of the senescence-associated β-galactosidase activity, hallmarks all known to be associated with the senescent phenotype. In this way, senescent melanoma cells acquired sensitivity to the BCL2 family inhibitor ABT263 [69].

Schepers et al. developed a suicide switch system to eliminate the undesirable proliferating cells, allowing the genome-wide CRISPR screening only in growth-arrested subpopulations. Using this system, several autophagy-related proteins were identified as targets for senescence induction in A549 lung cancer cells. The suicide switch system consists of a construct that contains an inducible caspase 9 system (iCasp9) driven by the Ki-67 promoter. The iCasp9 contains the intracellular portion of the human caspase 9 protein, fused to a drug-binding site responsible for the chemical induction of caspase 9 homodimers dimerization. The subsequent addition of the drug AP20187 leads to dimerization of the caspase 9 homodimers, resulting in cellular apoptosis and elimination of the proliferating cells. Then, the growth-arrested subpopulations, identified using specific cytofluorimetric markers, were infected with the Brunello CRISPR library, a library characterized by sgRNAs that target genes known to be associated with senescence. The genome-scale CRISPR screening led to identify *ATG9A*, *RB1CC1*, *ATG101* and *RAB14* as possible targets for senescence-inducing therapies. As a matter of fact, knocking out of these four genes in the proliferating cells induces tumor cell senescence as demonstrated by the expression of the typical senescence markers [70].

Cervera et al. used CRISPR/Cas9 genome editing technology to cause permanent inactivation of an important oncogene that drives Ewing sarcoma pathogenesis, inducing a senescence phenotype that could be exploited therapeutically. Ewing sarcoma is an aggressive bone tumor arising in children and young adults characterized by chromosomal translocations that give rise to fusion proteins governing tumorigenesis. The most frequent one, present in 85% of all cases, is the fusion of the protein’s genes *EWSR1*–*FLI1*. In particular, the N-terminal region of the *EWSR1* gene is fused to the C-terminal region of the transcription factor *FLI1* [71]. As an aberrant transcription factor, EWSR1–FLI1 establishes a specific transcriptional program that promotes cell proliferation and blocks cell differentiation. In this study a guide RNA that specifically and efficiently targets the *EWSR1*–*FLI1* gene was identified, inducing its genetic inactivation that in turns causes cell cycle arrest and senescent phenotype of the A673 Ewing sarcoma cells [72].

Over the years, CRISPR-Cas 9 has been also employed to better understand the molecular mechanisms by which senescence-inducing therapies arrest cell-cycle in cancer cells and the genes involved in conferring drug resistance or lack of response to these treatments. Carpintero-Fernández et al. used CRISPR-Cas9 technology to identify genes that participate in the arrest of proliferation induced by CDK4/6 inhibitors. It is known that CDK4/6 inhibitors (as Palpociclib, Abemaciclib) induce senescence and reduce tumor growth in breast cancer patients [73]. However, genes regulating senescence in this context are still unknown, limiting their antitumor activity. Using a human genome-wide CRISPR/Cas9 library it has been identified that loss-of-function of the coagulation factor IX (*F9*) gene prevents the cell cycle arrest and senescent-like phenotype induced by Palbociclib in MCF7 breast cancer. These results were confirmed using an alternative CDK4/6 inhibitor, Abemaciclib in another breast cancer cell line, T47D, where it was seen that also the downregulation of *F9* prevented the induction of senescence. The subsequent treatment with a recombinant *F9* protein was sufficient to induce a cell cycle arrest and senescence-like state in MCF7 and T47D tumor cells [74].

### 2.2. Mutations, Genomes and Beyond

Here, instead, we focused on those hallmarks that describe what does happen inside the nuclei, such as mutations of the sequence or rearrangements of the structure of the genome, which were reported to contribute to cancer progression.

#### 2.2.1. Accumulate Genomic Mutations

Cells are exposed to different amounts of mutagens for definite amounts of time on a daily basis, and the occurrence of significant mutagenic events is dramatically mitigated by several repair mechanisms, which either directly intervene on the genome to fix the damage or trigger apoptosis. All these steps strongly reduce the likelihood of allowing a damaged cell to transform into a cancer cell, therefore, to develop neoplasms. Most of these mechanisms are strongly interconnected to cell cycle control: indeed, *TP53*, *RB1* and *LKB1* represent the most important checkpoints during all these phases that lead to mitosis, and their veto means cell death. As we previously described CRISPR works that focused on such proteins, we summarize in Table 2 the results of those works that, recapitulating a particular mutation in a particular gene, demonstrated its driving or supporting role in the study of oncogenesis.

#### 2.2.2. Non-Mutational Epigenetics Reprogramming

Non-mutational epigenetic regulation of gene expression is an important process in embryonic development, organogenesis and differentiation. It recently emerged that reprogramming epigenetic events play a pivotal role in cancer development and progression. In particular, alterations of those genes involved in gene expression regulation are increasingly associated with tumor hallmarks. A cancer cell genome can be reprogrammed and tumor epigenome can be modified due to anomalous traits of the tumor microenvironment, principally referring to hypoxia-mediated epigenetic regulation and epithelial-to-mesenchymal transition (EMT). Additionally, epigenomic heterogeneity, from which only the most eligible cells will establish malignant progression, has been observed through analyses of DNA methylation profile, histone modification, chromatin accessibility, RNA translation and post-transcriptional modifications [11].

Considering the quite poor knowledge of cancer epigenetics, the CRISPR technique represents a useful tool to investigate epigenetic modifications’ role in cancer malignant progression and to find new possible targets for cancer therapy.

CRISPR technology has facilitated the generation of CRISPR/Cas9 knockout libraries, used to target epigenetic modifiers and to discover new chromatin-modifying genes responsible for different tumor fitness, like triple-negative breast and prostate cancer [82].

In renal cell carcinoma (RCC), atypical epigenetic reprogramming has been identified as a tumor hallmark. CRISPR/Cas9 screen has been employed to recognize and potentially target abnormal epigenetic targets. Through this method, for example, Jumonji domain-containing 6 (*JMJD6*) gene has been identified as an epigenetic vulnerability in RCC, leading to tumor progression via oncogenic transcriptome alteration. This result was confirmed by CRISPR/Cas9 *JMJD6*-KO generation of RCC cells. In this way, it has been recognized as both a cancer predictive biomarker and a therapeutic target [83].

The CRISPR screening approach has been used in different types of tumors in order to analyze cancer epigenetic regulation and identify new therapeutic opportunities. In Ewing sarcoma (EwS) cell lines, CRISPR/Cas9 high throughput screen has been performed in order to examine epigenetic alterations that alter genetic expression [84]. In murine pancreatic ductal adenocarcinoma (PDAC) cells, CRISPR/Cas9 screen has been essential to identify epigenetic modifiers with a crucial function in tumor plasticity [85]. CRISPR screen has also been used to study the key role of epigenetic regulation in intrinsic or acquired chemoresistance. In diffuse intrinsic pontine glioma (DIPG) cells, histone demethylase KDM1A has been identified as a key chromatin regulator, with an important function in DIPG sensitivity to histone deacetylase inhibitors [86]. In multiple myeloma cell lines, hepatoma-derived growth factor 2 (*HRP2*) has been distinguished as a principal chemosensitivity regulator through transcriptional events reprogramming [87]. In EGFR mutant non-small cell lung cancer (NSCLC), where genetic and epigenetic mechanisms can lead to drug resistance and cause EGFR tyrosine kinase inhibitors (TKIs) failure, FGFR1 has been selected as the best target to re-sensitize cells to EGFR inhibition and contrast EMT-related adaptive resistance [88].

In order to study DNA methylations as markers of prostate carcinogenesis, CRISPR/Cas9 knock-out of tumor-suppressor *TET2*, which is involved in DNA demethylation and is usually poorly expressed in prostate cancer cells, has been carried out and helped to identify highly altered genes in prostate cancer. This analysis resulted in a wide epigenetic silencing that could mimic prostate cancer cells epigenetic profile [89].

CRISPR/Cas9 knock-out of the histone modifier Ring1b in murine pancreatic cancer cells has enabled the study of epigenetic alteration dynamics that lead to pancreatic carcinogenesis [90]. In giant cell tumor of bone (GCT), CRISPR/Cas9 was used to generate G34W-mutant *H3F3A*-KI clones in order to determine the role of this oncohistone on tumorigenesis. This technique contributed to the recognition of the G34W-mediated epigenetic remodeling, investigating the role of this mutation on chromatin remodeling and neoplastic maintenance [91].

Further, CRISPR/Cas9 technologies have been used to create overexpressing or knock-out cells of a chromatin remodeling enzyme, the lymphoid-specific helicase HELLS. HELLS has then been recognized as an essential epigenetic driver in hepatocellular carcinoma cells, epigenetically suppressing different tumor suppressors [92].

Furthermore, CRISPR/Cas9 has been a useful tool for the investigation of epigenetic alterations connected to epithelial–mesenchymal plasticity (EMP). Gene expression modulation through CRISPR approaches, such as CRISPR-mediated knockout or CRISPR activation of crucial genes, can investigate the effects of aberrant transcription factors expression and discover epigenetic regulators of EMT, which are events strictly connected to progression, metastasis and recurrence of different tumor types [93,94].

Finally, CRISPR activation strategy has been employed in human breast cancer cell lines for the re-activation of FOXP3, together with CRISPR interference approach used to silence *XIST* in the same cells. This double system led to a lower cell growth and DNA methylation, altering both transcription and epigenetic modifications of the targeted loci [95].

### 2.3. Energy and Motion

In this section we gathered all those hallmarks, mostly metabolic, associated to the production and employment of energy, namely the regulation of the metabolism, the ability to switch phenotype and the acquisition of motility.

#### 2.3.1. Reprogram Cellular Metabolism

To adjust to the surrounding environment or because of upstream signaling events, cancer cells are known to be able to rewire their metabolism. This has relevance not only due to the emergence of new metabolic abilities or to a fitter status that allow these cells to thrive in conditions that would otherwise induce senescence or apoptosis to healthy cells, but also because of pharmacologic implications. Michl et al. ran a CRISPR screening on colorectal cancer cells SW1222, the most acid-tolerant, SW480, with an intermediate tolerance, and COLO320DM, the most acid-sensitive, at physiologic and acidic pH to detect gene expression alterations that would grant those cells resistance to such low pH. The top entries were *NDUFS1*, *NDUFS2* and *NDUFA1* subunits of mitochondrial complex I, *COX8A* subunit of mitochondrial complex IV, and *IBA57* and *NFU1* subunit of iron-sulphur cluster related to the functionality of complex I itself. These genes gave the clue that under acidic conditions, cancer cells have to rely on oxidative phosphorylation. On the other hand, *ALDOA* inactivation induced the greatest cell death at physiologic pH, indicating that tumor cells strongly rely on glycolysis in these conditions [96].

Analogously, Mennuni et al. performed a CRISPR screening while treating RKO colon cancer cells with IMT1, a mitochondrial DNA transcription inhibitor. Resistance to such very specific drugs was conferred by VHL and mTORC1, indeed their inhibition resulted in an increased sensitivity to the drug [97].

Still using CRISPR screening, Li et al. identified hexokinase 2 (*HK2*) as a self-renewal promoter of liver cancer stem cells population of HUH7 cell line. The mechanism involves pushing glycolysis to produce and accumulate acetyl-CoA, which epigenetically can upregulate *ACSL4* that regulates both beta-oxidation pathway and fatty acid synthesis [98].

In addition, CRISPR/Cas9 has been used to generate the knock-out of the B subunit of succinate dehydrogenase enzyme (*SDHB*) in hPheo1 cell line, a progenitor cell line derived from a human pheochromocytoma tumor allowing to observe and analyze the consequences in proliferation, cellular adhesion, mitochondrial respiration, glycolysis and glutaminolysis [99].

#### 2.3.2. Unlocking Phenotypic Plasticity

Cellular plasticity is defined as the ability of cells to assume a wide range of distinct cellular phenotypes via changes in gene expression patterns. Normally, cellular plasticity is a fundamental feature for biological processes like normal development, regeneration and tissue homeostasis, but it is also a typical characteristic of fully formed tumors. Thus, cancer cells are more plastic than terminally differentiated healthy cells and this advantage allows them to undergo functional adaptations which sustain carcinogenesis as well as intratumoral heterogeneity and therapy resistance [100]. In support of this hypothesis, Hanahan et al. incorporated phenotypic plasticity as an emerging hallmark in the classical view of cancer conceptualization, and divided it in three main manifestations: dedifferentiation from mature to progenitor states, blocked (terminal) differentiation from progenitor cell states and transdifferentiation into different cell lineages [11].

In the last few years, CRISPR/Cas9 technology has facilitated the study of these new tumor characteristics. From the transdifferentiation point of view, pancreatic cancer is an optimal example because these cancer cells originate from reprogrammed acinar cells, which undergo a transient transformation losing acinar phenotype and displaying a ductal morphology of a progenitor-like cell type, making acinar-to-ductal transdifferentiation a key phase in the initiation of pancreatic cancer. In this regard, Yasunaga et al. reported that the loss of alpha amylase (*AMY2*), a pancreatic enzyme produced in acinar cells, contributes to pancreatic cancer development via acinar-to-ductal metaplasia through autophagy. These results were obtained thanks to CRISPR/Cas9 system by studying *AMY2*-KO cells autophagy upon treatment with rapamycin and caerulein and comparing autophagy marker LC3-II expression to that of *ATG12*-KO, which were generated to deplete autophagy from MIA PaCa2 and AR42J cells [101].

Moreover, to define cellular identity during pancreatic carcinogenesis, the epigenetic remodeling may be evaluated. It has been reported that knocking out *Ring1b* lead to the catalysis of histone modification H2AK119ub, namely the epigenetic silencing of acinar regulatory transcription factors: this editing promoted mouse pancreatic tumor cells reprogramming towards a less aggressive phenotype. Thus, *Ring1b* depletion resulted in a decrease in acinar-to-ductal metaplasia (ADM) because acinar cells were maintained in a differentiated state [90].

Another manifestation of phenotypic plasticity is the dedifferentiation process, which plays a crucial role in several tumors including melanoma. As a matter of fact, melanoma cells undergo a phenotypic switch due to loss of melanocytes specific gene expression and increase of mesenchymal markers leading to the acquisition of an aggressive undifferentiated phenotype. Among these genes, downregulation of the transcription factor MITF appears as a master regulator of melanocytes dedifferentiation. Thus, in a recent study, the generation of *SOX10* knockout (*SOX10*-KO) in MITF methylated melanoma cell has shown that *SOX10*-KO cells revert to a pre-neural crest state characterized by the downregulation of *SOX2*, *SOX5*, *SOX8*, *SNAI2* and increased expression of *SOX9* [102].

Another escamotage used by tumor cells to acquire a more plastic phenotype is that of blocked differentiation, namely inducing well-differentiated cells to dedifferentiate into progenitors. A great example is that of rhabdomyosarcoma (RMS), in which the blockade of myogenic differentiation program is essential to induce tumor progression. Phelps et al., using CRISPR/Cas9 genome-editing technology, have discovered that the Nuclear Receptor Corepressor (NCOR)/Histone deacetylases (HDAC3) complex blocks myogenic differentiation in RMS. Indeed, *HDAC3*-KO in RMS cells result in a strong decrease in tumor growth due to the activation of terminal myogenic phenotype differentiation [103].

Moreover, in breast tumors, the overexpression of Forkhead box (*FOXM1*) transcription factor is critical for the phenotypic plasticity of said cells. Normally, FoxM1 associates with the CREB-binding protein (CBP) to activate gene transcription, however, it also exerts a transcriptional repressor function by associating with the Rb protein. Recently, Kopanja et al., using CRISPR–Cas9 engineering, have generated a KO mouse model expressing FoxM1 point mutations that block the binding to Rb while maintaining its ability to bind CBP. Their results have shown that the loss of FoxM1/Rb interaction induced an expansion of the differentiated alveolar tumor cells, indicating that FoxM1/Rb is a key player in the evolution of metastatic cells by suppressing differentiation genes, including Gata3 [104].

Cancer cell plasticity, also referred as epithelial-to mesenchymal transition (EMT), is a dynamic and reversible cellular program by which neoplastic epithelial cells transit into a completely mesenchymal state that confers them stem cell-like properties, increases their motility and invasive capacity [105]. Nowadays, it has become clear that the highest phenotypic plasticity is shown by cancer cells in hybrid, intermediate, or incomplete states characterized by co-expression of epithelial and mesenchymal markers. However, much remains to be investigated regarding the specific programs that govern phenotype plasticity in this partial EMT-like state [106].

At present, CRISPR technology was used in several studies focused on this topic, especially for high-throughput screening of cancer-related genes. For example, in melanoma, gain-of-function CRISPR screens have identified *SMAD3*, *BIRC3*, and *SLC9A5* as main drivers of resistance to BRAF inhibitors, but only the upregulation of *SMAD3* transcriptional activity induces a mesenchymal-like phenotype as well as BRAFi resistance [107]. Serresi et al. have identified those factors required for the proper regulation of epithelial-mesenchymal interconversion using a large scale-CRISPR interference screen by a phenotypic CRISPRi strategy, they found both known and unknown EMT regulators, the latter including *CNKSR2*, which is a new driver in a RAS–dependent signaling directly linked to EMT and chromatin regulation [108].

#### 2.3.3. Activate Invasion and Metastasis

Metastasis is a multistep and multifactorial process, which includes dissociation of tumor cells from the primary site, anchorage-independent growth, apoptosis evasion, cell migration, invasion of surrounding tissues, intravasation into the circulation, extravasation at metastatic site, survival at secondary site, and finally formation and eventually proliferation of secondary tumors.

Metastatic progression is typical of colorectal cancer (CRC); indeed, it is one of the tumors with more cancer-related deaths correlated to metastasization. Nevertheless, the molecular mechanisms involved in the first steps of its metastatic dissemination are mostly unknown [109]. In a recent report, Huebner et al., have identified activating transcription factor 2 (*ATF2*) as a potential new therapeutic target in CRC. In this study, using CRISPR/Cas9-mediated *ATF2*-KO cells, they discovered that ATF2 acts as a tumor suppressor by inhibiting the cancer driver trophoblast cell surface antigen 2 (*TROP2*), which is associated with cell de-adhesion and cell migration without triggering EMT. Indeed, *ATF2*-KO clones revealed an upregulation of *TROP2* expression leading to increased invasion in vivo in a mouse model as well as in chicken xenograft models [25]. During metastatic progression, which critically depends on the dynamic interplay between tumor cells and TME, cancer cells are more susceptible to acquire resistance to detachment-induced apoptosis, also known as anoikis. However, little is known about the acquisition of the ability to survive under “anchorage-independent” growth conditions. Zhang et al., thanks to genome wide CRISPR/Cas9 knockout screen, have identified critical drivers of anoikis, using ovarian cancer cell lines in ultra-low attachment conditions as a model that mimics the situation in which cancer cells shed from the primary attachment. They demonstrated that knockout of Protein-L-Isoaspartate (D-Aspartate) O-methyltransferase (*PCMT1*), the main driver in anoikis resistance that they identified, caused an increased apoptosis of SKOV3 cells in response to detachment from the ECM, which is correlated with a decreased tumorigenesis and metastasization in the xenograft model [110].

In breast cancer, Wang et al. investigated the role of anillin, a unique scaffolding protein regulating major cytoskeletal structures, and playing a key role in breast cancer metastatization. Thus, using both *ANLN*-KO clones, derived from highly metastatic breast cell line (MDA-MB-231 and BT549) and anillin-overexpressing clones from poorly invasive MCF10AneoT cells, they have shown that anillin is necessary and sufficient to induce breast cancer anchorage-independent growth, motility and metastasis in vitro [111]. Moreover, it has been reported that the activation of bone morphogenetic protein (BMP) signaling and upregulated levels of BMP-antagonists, such as gremlin1, correlated with breast cancer progression and metastasis. Neckmann et al., have shown that expression of *GREM1* is associated with extracellular matrix organization, formation, biosynthesis of collagen and its expression, predicted poor patient prognosis in estrogen receptor negative breast cancer. In line with these results, using CRISPR/Cas9 gene editing to generate *GREM1*-KO in breast cancer cells which metastasizes to the lungs (66cl4 cells), they have reported that depletion of gremlin1 induces the formation of smaller primary tumors and impairs metastasis formation in the lungs in immunocompromised nude mice [112].

Al-Mulhim et al. have evaluated the efficacy of the CRISPR/Cas9-edited breast cancer cells to control the invasion, metastasis of mammary gland tumor in rats. They have focused their attention on two key players of breast cancer progression: *CDH1* gene which encoded for epithelial cadherin (E-cadherin) and *CDK11*, which is a member of the serine/threonine protein kinase family that plays crucial roles in tumor cell proliferation and growth by controlling cell-cycle. It is known that in breast cancer E-cadherin is inactivated and is also known that the loss of *CDH1* function induces EMT, causing dysregulation of cell–cell adhesion and anoikis resistance [113,114]. Thus, Al-Mulhim et al. have demonstrated that the subcutaneously inoculation in rats of MCF-7 breast adenocarcinoma cells, engineered by CRISPR/Cas9 to either activate *CDH1* or knock out *CDK1*, resulted in minimal tumor cells infiltration and invasion, indicating that dual targeting could be a better mechanism to inhibit metastasis of breast cancer in vivo [115].

Another new metastatic target identified by CRISPR/Cas9 is Urokinase plasminogen activator (uPA) receptor (uPAR, gene symbol: *PLAUR*). uPAR is a glycosylphosphatidylinositol-anchored glycoprotein which has emerged as a potential regulator of remodeling of extracellular matrix, cell adhesion, cell migration, proliferation, differentiation, and cell survival in different physiologic and pathologic contexts. In a recent report by performing two different *PLAUR*-KO cancer cell lines by CRISPR/Cas9 system, it has been shown that depletion of uPAR causes the inhibition of cell proliferation, migration and invasion, indicating that uPAR expression in this model promotes the proliferation, metastasis, and invasion of cancer cells [116].

### 2.4. Communication with the Outer World

In this latter section, we put a spotlight on the complexity of the disease, pointing at those hallmarks that concern the communication among cancer cells, immune cells and other stromal cells in the neoplastic milieu.

#### 2.4.1. Supporting the Inflammatory Response

It is now accepted that two different pathways could explain cancer-related inflammation: within the extrinsic pathway, chronic inflammatory conditions increase cancer risk, while in the framework of the intrinsic pathway, genetic alterations may cause inflammation and the development of neoplasia by promoting the formation of an inflammatory tumor microenvironment [117,118]. It is therefore clear that in the last few years, several studies have focused on this topic in order to identify new target molecules that could lead to improved diagnosis and treatment [119].

Currently, thanks to CRISPR/Cas9, several KO cell lines have been obtained also to better understand cancer-related inflammation.

Watanabe et al. have evaluated the influence of p53 function on sporadic colorectal neoplasm under chronic inflammation to obtain a long-term inflammation model of *TP53*-mutated LS174T cells. Thanks to this *TP53* mutation model they established that the acquisition of more malignant phenotypes, even in sporadic colorectal cancer cells under chronic inflammation, was mediated by p53 [120].

As a matter of fact, many studies have focused on pro-inflammatory signaling pathways in the colon because chronic inflammation is a predisposing condition for colorectal cancer, and CRISPR/Cas9 is helping to find mechanisms that suppress inflammation, which is critical for developing therapeutic interventions. Means et al. have explored the roles of transforming growth factor beta (TGFβ) family signaling through SMAD4 in colonic epithelial cells by knocking out *SMAD4* gene in human colorectal cancer cells. The authors observed that the loss of *SMAD4* alone increased the expression of a broad range of inflammatory mediators and is sufficient to initiate inflammation-driven carcinogenesis in the colon [121].

Shi et al. applied the CRISPR/Cas9 system to obtain Isochorismatase domain-containing protein 1-knock out (*ISOC1*-KO) in lung cancer cells. Thus, thanks to co-immunoprecipitation combined with mass spectrometry and RNA sequencing of these KO cell lines they showed that *ISOC1* exhibited a tumor-promoting function in lung cancer by interacting with DNA damage repair pathways and mediating inflammation-related signaling pathways [122].

UV light exposure is the major environmental risk factor for melanoma development, by inducing oncogenic mutations as well as promoting an inflammatory microenvironment that supports tumorigenesis. Mengoni et al. reported that aryl hydrocarbon receptor (*AHR*), a ligand binding-transcription factor, is enhanced by inflammatory mediators and it increases inflammation-induced dedifferentiation in human and mouse melanoma cells. These results have been obtained with a CRISPR/Cas9-mediated disruption of the *AHR* gene in melanoma cells, which showed a decreased tendency for inflammation-induced dedifferentiation. When *AHR*-KO melanoma cells have been transplanted into immunocompetent mice, local growth and metastatic dissemination were decreased, suggesting a functional role of AHR in cancer induced-inflammation [123].

The tumor microenvironment (TME) contributes to tumor inflammation; TME is a complex ecology of heterogeneous cell populations in which tumor associated macrophages (TAMs) represent the predominant elements, showing a dominant role as orchestrators of cancer related inflammation [124].

Recently, Wang et al., analyzing cancer cell and TAMs interaction in breast cancer, discovered that Lysosome associated membrane protein type 2A (*LAMP2A*), which contributes to chaperone-mediated autophagy, is a new potential candidate in TAM-targeting tumor immunotherapy because its inactivation induced by CRISPR/Cas9 prevents TAMs activation, tumor growth and restores immune-environment in tumor milieu [125].

#### 2.4.2. Enabling Immune Evasion

The immune system acts as a constantly active barrier against tumor development and progression by limiting formation of above 80% of tumors of nonviral etiology. Then, enabling immune evasion is regarded as one of the most well-established hallmarks of cancer. According to the theory of immune surveillance, cells and tissues are continuously monitored by the immune system which, as an alarm bell, eventually recognizes and eliminates the vast majority of pre-cancerous cells and nascent tumors. Therefore, visible tumors have managed to avoid or have been able to limit this type of detection [10]. The importance of the immune system in cancer genesis and development is supported by the fact that certain types of cancers arise more frequently and/or grow more rapidly in immunocompromised individuals and immunodeficient mice in comparison to respective immunocompetent controls [126,127].

Moreover, the pivotal role of the immune system in cancer is justified by the fact that the presence of immune responses in the tumor microenvironment defines the response to treatments: patients with tumors that are heavily infiltrated with CTLs and NK cells have a better prognosis than those lacking immune infiltration [128]. What is known is that, in a certain moment of tumor progression, cancer cells manage to escape immune detection by activating negative regulatory checkpoints thus inhibiting immune responses. Among the numerous checkpoints, the most studied are cytotoxic T lymphocyte protein 4 (CTLA4) and programmed cell death protein 1 (PD-1), the former controlling T cell activation, the latter inhibiting T cell proliferation by inducing cell apoptosis [129].

CRISPR/Cas9 gene editing technology has received increasing attention in exploring tumor immune mechanisms and biomarker screening. Moreover, CRISPR/Cas9 has also shown great potential in improving clinical trials and therapies that act by reactivating the immune responses in the tumor microenvironment.

PD-1/PD-L1 has been identified as one of the most important negative immunomodulatory molecular axes that promotes immune evasion of tumor cells. Immunotherapy strategies that target in a direct or indirect way PD-1/PD-L1 interaction have been developed in order to reactivate the immune system in tumor microenvironments. In this context, CRISPR/Cas9 gene editing technology has been used to successfully edit the axis of PD-1/PD-L1, knocking out *PD-1* through different delivery methods in different cell lines, including primary T cells, CTLs, NK cells and B cells with the purpose of enhancing the antitumor immunity [130,131].

PD-1 and/or PD-L1 are highly expressed on the surface of various malignant tumors, a fact that is associated with poor prognosis of patients. CRISPR/Cas9 has been also employed to edit *PD-L1* directly in tumor cells. Its knockout promotes tumor antigen presentation, immune cell proliferation and cytotoxicity in the tumor microenvironment, improving tumor chemotherapy resistance. PD-L1 expression level reduction enhances the recruitment of different types of immune cells, including CD4 T cells, CD8 T cells, NK cells and CD11c+ M1-matured macrophages, while decreasing regulatory T cells in the tumor microenvironment. Xu et al. reviewed and summarized the role of CRISPR/Cas9 technology- associated PD-1/PD-L1 editing in tumor immunity in different types of tumors, including leukemia, multiple myeloma, breast cancer, hepatocarcinoma, lung cancer, melanoma, cervical and ovarian cancer [132].

Wang et al. found that CRISPR/Cas9-mediated knockout of *SHP2*, a tyrosine phosphatase involved in the regulation of immune cells signaling, in the human ovarian carcinoma cell line SHP09 inhibited its protein activity and enhanced tumor intrinsic IFN-γ signaling. This in turn resulted in increased expression of its downstream targets, including chemoattractant cytokine release and cytotoxic T cell recruitment, as well as increased expression of MHC class I and PD-L1 on tumor cells surface. Furthermore, SHP2 inhibition promoted T cell proliferation and reduced the differentiation and suppressive function of immunosuppressive myeloid cells in the tumor microenvironment. This study revealed that SHP2 may become a promising target for cancer immunotherapy leading to favorable changes in the tumor microenvironment and controlling cancer progression [133].

Yang et al. have found that successful nuclear localization of CRISPR/Cas9 ensured efficient destruction of both *PD-L1* and *PTPN2* (protein tyrosine phosphatase N2) in melanoma cell line B16-F10. *PD-L1* downregulation in tumor cells disrupts PD-1/PD-L1 interaction attenuating the immunosurveillance evasion and enhancing adaptive immunity by spurring potent immune T cell responses. Deletion of *PTPN2* can modulate the inhibition of the JAK/STAT pathway and promote tumor susceptibility to CD8+ T cells dependent on IFN-γ, thus further amplifying the adaptive immune response. This study provided a promising alternative to current PD-L1 immune checkpoint blockade monotherapy [134].

Finally, the CRISPR/Cas9 technology has provided a multifunctional and efficient method for the transformation of engineered T cells. In particular, the use of CRISPR/Cas9-mediated gene editing has been explored to enhance T cell effector functions, to prevent T cell dysfunction through interfering with inhibitory receptor signaling and to redirect T cell antigen specificity. These approaches, all reviewed by Heeren et al., have led to the development of genetically engineered T cells that have the ability to kill tumor cells triggering significant therapeutic effects. Nevertheless, the current CRISPR/Cas9 methods are suitable for the manipulation of small numbers of cells, the editing of the large numbers of cells required in clinic for patient treatment might still be sub-optimal and consequently novel CRISPR/Cas9 approaches should be developed [135].

In addition, CRISPR activation strategy has been used in different tumor cell types, leading to a major presentation of tumor antigens. This approach could permit to increase anti-tumor immune responses overcoming the frequent antigen loss in tumoral cells [136].

#### 2.4.3. Inducing the Formation of New Vessels

Sustained angiogenesis is an essential prerequisite to the clonal expansion of tumoral cells; therefore, it is one of the hallmarks of cancer that are necessary for the development of solid and macroscopic tumors. The ability to induce the formation of new vessels from pre-existing ones seems to be acquired during tumor progression via the so-called multi-step “angiogenic switch” starting from a status of vascular quiescence. Tumors activate the angiogenic switch by changing the balance between angiogenesis inducers and inhibitor counterparts via alteration of gene transcription. Indeed, thanks to the genomic instability, tumor cells increase expression of angiogenesis-initiating genes, including VEGF and/or FGFs, and downregulate expression of endogenous inhibitors such as thrombospondin-1 and interferon-β [137].

The regulation of the angiogenesis pathway represents an important tool to control tumor progression and spreading. To date, a lot of drugs have been developed as therapeutic agents against angiogenesis and are currently used in cancer therapy [138]. Over time, though, resistance to such molecules developed and in more and more patients they have become ineffective. To better understand the role of genes involved in tumor angiogenesis, CRISPR/Cas9 technology has been successfully used as well. Hariprabu et al. reviewed the current applications of CRISPR/Cas9 technology in tumor angiogenesis research for the purpose of cancer treatment, with a focus on the possibility to create KO models to study growth factors, cytokines, kinases and integrins involved in the angiogenesis switch as possible anti-angiogenic targets [139].

Zhu et al. used CRISPR/Cas9 gene-editing technology to knockout the *EGFL6* gene in the ovarian cancer cell line SKOV3 by designing a specific guide RNA targeting its exons. EGFL6 is a protein known to be highly expressed in ovarian cancer and proposed to play a key role in promoting tumor angiogenesis. Its knockout markedly inhibited the proliferation, migration, and invasion of SKOV3 cells, as well as promoted apoptosis of tumor cells by downregulating FGF-2/PDGFB signaling pathway [140].

Following the same experimental trail, Chen et al. used CRISPR/Cas9 gene-editing technology to prove that global microRNA depletion, induced by knocking out *DICER1*, suppresses tumor angiogenesis in a non-small cell lung cancer (NSCLC) model. The miRNA-deficient tumors resulted highly hypoxic but poorly vascularized. As a result of the microRNA deficiency, angiogenesis genes were significantly downregulated. Moreover, they found that the reduced angiogenic capacity is primarily mediated by derepression of FIH1 (factor inhibiting HIF-1), which inhibits HIF transcriptional activity thereby suppressing the response to hypoxia. Knocking out *FIH1* using CRISPR/Cas9-mediated genome engineering reversed the phenotypes of microRNA-deficient cells and increased HIF transcriptional activity, VEGF production and tumor angiogenesis. Then, by using multiplexed CRISPR/Cas9, they deleted 3′-untranslated regions (UTRs) in FIH1 that contain microRNA-binding sites. These specific regions deletion caused FIH1 protein derepression and hypoxia response repression. All together, these data suggest that microRNAs promote tumor responses to hypoxia and angiogenesis by repressing FIH1 [141].

Tsai et al. used CRISPR/Cas9-mediated genome editing to investigate the oncogenic role of KDR, the predominant mediator of VEGF-induced angiogenesis, in SW579 squamous advanced thyroid cancer cell line. The *KDR*-KO significantly reduced SW579 sprouts formation in vitro, confirming that including selective targeting of KDR to anti-cancer therapy to suppress the VEGF/VEGFR axis in patients with clinical advanced thyroid cancers might increase its efficacy [142].

#### 2.4.4. Polymorphic Microbiomes

Several studies carried out in human and in mouse models with respect to cancer have revealed that particular microorganisms, principally bacteria, can have either protective and/or deleterious effects on cancer development, malignant progression, response to therapy and eventually drug resistance.

In particular, the gut microbiome has been the first identified in this new frontier, allowing the growing appreciation of the importance of polymorphically variable microbiomes in the acquisition of hallmark capabilities of cancer. The true question is if this appreciation is enough to regard polymorphic microbiomes as one of the new enabling characteristics that must be acquired by cells to become malignant [11].

Considering its importance in the physiological function of the large intestine, the gut microbiome has been associated with the susceptibility, development, pathogenesis, and progression of colon cancer [143]. A lot of evidence has revealed that there are both cancer-protective and tumor-promoting microbiomes, according to different types of bacterial species [144]. On one side, bacteria have been reported to produce and secrete ligand-mimetics that, by binding receptors expressed on the surface of colonic epithelial cells, can stimulate epithelial proliferation and survival, contributing to the transformation of healthy cells into neoplastic cells [145]. On the other hand, bacteria can trigger both innate and adaptive immune responses leading to the secretion of cytokines and chemokines that constitute a protection against tumorigenesis [146].

CRISPR/Cas-9 technology has not been employed yet in the comprehension of this complex cancer hallmark. In the future, it would be interesting to employ this to better understand the molecular mechanisms by which polymorphic microbiomes influence tumor pathogenesis and progression.

## 3. Discussion and Future Perspectives

Ten years ago, the CRISPR breakthrough was perceived as representative of a disruption of the past, allowing the rise of more affordable and precise tools for genome editing. By studying Cas enzymes from different bacteria and pursuing their engineering, this technology has reached, in such a short time, a huge degree of versatility, allowing scientists worldwide to push the throttle and study several biological processes in-depth. These advantages though are counterbalanced by its low efficiency and its dependency on clonality, as well as by the possible occurrence of off targets. Indeed, due to the heterogenic nature of cancer bulks, what could occur can be that during the selection process of CRISPR-affected clones, cells with different drug-sensitivity or populations with early or late differentiation status might be lost, thus making it more complex to obtain a very well characterized and representative model of the original tumor bulk.

For what concerns cancer, several groups focused on well-established mechanisms to elucidate further important details, whilst others pursued more neglected paths; in general, great efforts were concentrated both in vitro and in vivo systems.

The aim of this review was to revise the literature, looking for in vitro applications of CRISPR, highlighting particularly elaborate strategies, and summarizing the main discoveries with a particular impact on cancer research. So far, CRISPR is already representing an extremely powerful research tool for understanding more finely more and more processes that underlie oncogenesis, and facilitating the exploration of the impact that single pathways or molecules have on one another, in the next decade will surely boost our general knowledge and ability to tackle cancer more efficiently.

It is worth mentioning that, given the vastness of this research field, we deliberately decided to focus on summarizing only the in vitro side of CRISPR-related cancer research with no intention to belittle in vivo works, whose results actually have a greater impact on drug development and clinical application of CRISPR-based technologies. On the other hand, we believe that gathering all the advances pursued in vitro could provide the rationale for translating these approaches to in vivo studies or even to clinics. Moreover, the generation of CRISPR-affected in vitro models and their subsequent employment in vivo has demonstrated to be extremely informative about the role of different genes.

In the near future, more studies are expected to push further the application of CRISPR for the direct comprehension of the mechanisms in vivo in more complex and translationally relevant models, as well as in the engineering of tools used in adoptive cell therapies that are gaining more and more importance in many fields ranging from cancer to degenerative diseases.

## Figures and Tables

**Figure 1 cancers-14-05746-f001:**
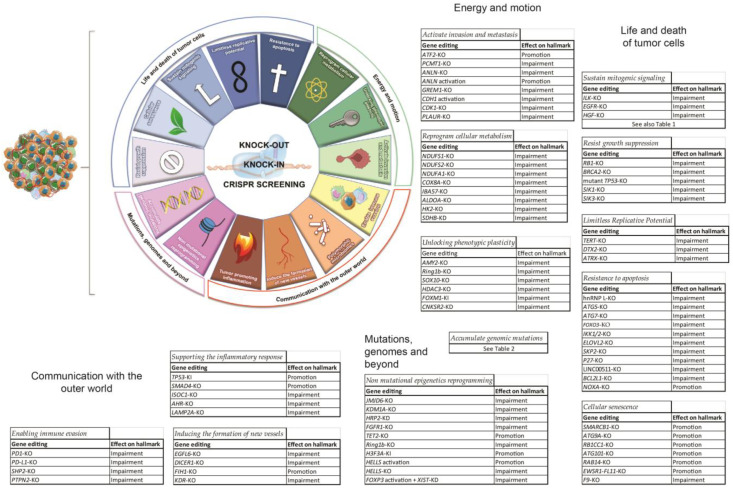
Classic representation of cancer hallmarks according to Hanahan’s latest review (2022). Surrounding the graphic, tables listing the genome editing targets described in the corresponding section and the effect they had in vitro according to the corresponding hallmark.

**Table 1 cancers-14-05746-t001:** CRISPR-targeted genes that impaired cell proliferation.

Model	District	Technique	Target	Suggested Mechanism (If Applicable)	Ref.
MCF-7	Breast	KO	*DCAF13*	Accumulation of PERP *	[12]
MOLM-13	AML	CRISPR screening	*KAT2A*	Inhibition of leukemogenic transcriptional programs, induction to differentiation leading to apoptosis	[13]
OCI-AML2					
OCI-AML3					
MDA-MB-231	Breast	Inducible KO	*RLIP*	Downregulation of surviving and Bcl-2; upregulation of Bim	[14]
MCF-7					
NCI-H460	NSCLC	CRISPR screening	*MDM2*	Removal of p53-inhibiting factor	[15]
A549					
OSRC	RCC	dCas13b-dependent methylation	*ZNF677*	-	[16]
CAK12					
MDA-MB-231	Breast	Cas9 knockdown	*B2AR*	Disruption of the B2AR-MOR interaction resulted in less aggressive phenotype	[17]
MDA-MB-468			*MOR*		
MDA-MB-231	Breast	Kinome-wide CRISPR screening	*PLK1*	-	[18]
MDA-MB-468					
T47D	Breast	KO	*HO-1*	-	[19]
HCT116	CRC	KO	*OLA1*	Downregulation of CA9 and HIF-1α	[20]
Lovo					
Kyse-30	Esophagus	CRISPR/dCas9	*ZNF154*	Targeted demethylation of ZNF154 promoter induced ZNF154 expression and inhibited proliferation	[21]
Kyse-140					
MCF-7	Breast	CRISPR screening	*ARID1A*	-	[22]
FaDu	HNSCC	KO	*SEC62*	-	[23]
PDX366	PDA	CRISPR screening	*ISL2*	-	[24]
HCT116	CRC	KO	*ATF2*	Inhibition of the cancer driver *TROP2*	[25]
HT29					
OCM1	UM	CRISPR screening	*GPS2*	Upregulation of oncogenic MAPK and PI3K-Akt pathways and downregulation of Slit/Robo pathway	[26]
MCF-7	Breast	KO	Linc-RoR (lncRNA)	Increase in the protein stability of DUSP7 decreasing ERK phosphorylation	[27]
AGS	GC	CRISPR screening	*METTL1*		[28]
Huh7	HCC	KO	*SMPDL3A*	Suppression of tumor proliferation and promotion of apoptosis through ERH	[29]
HepG2					
Neuro2a	Neuroblastoma	KO	*PLAUR*	p38 activation and decreased p53-mediated chemosensitivity	[30]
GIC	Glioblastoma	KO	*CD95*	Acquired resistance to CD95L-induced apoptosis	[31]

* p53 apoptosis effector related to *PMP22.*

**Table 2 cancers-14-05746-t002:** List of mutations knocked into different in vitro models.

Model	District	Target	Mutation	Effect	Ref.
HCT116	CRC	*KRAS*	G13D	Increasing zygosity of the mutant increased sensitivity to MAPK inhibitors	[75]
HeLa *	-	*DNMT3A*	K299I	The mutation altered the methylation pattern of the genome	[76]
H1975	NSCLC	*EGFR*	C797S	The mutation recapitulated the resistance to third generation TKis and showed upregulation of *AXL*	[77]
MCF-7	Breast	*FOXA1*	K295A	Permanent acetylation mimic of *FOXA1* in breast cancer	[78]
HEK293T *	-	*TRF1*	T273A/T358A	Inhibition of PI3K/Akt pathway	[79]
MCF-7	Breast	*ESR1*	D538G/Y537S	Increased sensitivity to ERD-148, new generation PROTAC	[80]
MCF-7	Breast	*ESR1*	D538G/Y537S	The mutations recapitulated the ligand independent ERα transcriptional activity, ligand-independent growth and endocrine resistance	[81]
T47D					

* Proof of concept studies.

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
