# Peer review of "Ten Years of CRISPRing Cancers In Vitro"

_cancers, 2022, doi:10.3390/cancers14235746_

Round 1

Reviewer 1 Report

Capoferri et al. reviewed the 10-years’ applications of CRISPR technology in cancer research, and specifically described them according to the categories of cancer hallmarks. Overall, this review is well written, however, I have some comments below.

1. Given the title “Ten years of CRISPRing cancers in vitro”, I believe it would be more interesting if the topic is not limited to in vitro, because CRISPR technology has been utilized into many aspects of cancer research, both in vitro and in vivo. The potential and significance of the application in in vivo or cancer treatment should be emphasized.

2. Since the subject of this review is associated with the application of CRISPR technology in cancer biology, the authors need include a summary/description of CRISPR technologies.

it would be best if the authors could provide a section describing the different CRISPR techs and summarize their applications in cancer research, e.g. using a figure or a table.

3. CRISPR technologies include not only CRISPR/Cas9 knockout, but also CRISPR interference, activation and base editing. Actually, all of these technologies have been applied into cancer research. However, little information besides CRISPR/Cas9 knockout was provided in this review.

4. “In this review we aim to report how this genome editing technique burst in the in vitro modeling of different aspects of tumor biology, its several declinations, and analyze the advantages and drawbacks of each of them.” However, I’m not able to find the descriptions of this statement, for example, how to burst, advantages and drawbacks. To discuss the burst, a timeline or an overview of the application of different CRISPR techs in cancer research is a good way.

5. Considering the length of writing, it would be clearer and more readable if a catalogue of all sections could be included at the front of this review, or a figure presenting the aspects discussed in this review.

6. The authors claimed that Table 1 gathers all the genes encoding intracellular proteins associated with proliferation. I doubt that all the related genes reported in last ten years are included.

7. Some sentences just mentioned the CRISPR/Cas9-mediated gene editing, but not indicated what kind of editing, knockout, knockdown, or activation?

8. Another hallmark of cancer is the heterogeneity. How CRISPR techs are applied for cancer research in this field? For example, the combination of CRISPR techs and single cell techs? As a hotspot of cancer research, it is more interesting if the authors could discuss it in a section.

Author Response

Response in the attched Cover letter

Reviewer 2 Report

The authors summarized some aspects of "Ten years of CRISPRing cancers in vitro'. In the past 10 years, there has been a lot of studies done using CRISPR technologies on multiple targets and cancer cell models. So the title needs to be more specific to be able to cover the area that has been discussed in the paper.  Also each section of discussing new pathways need a diagram for that pathway as in the current format, it is very hard for the reader to follow the long text sections and the context. 

Author Response

Response in the attched Cover letter

Round 2

Reviewer 1 Report

The authors have addressed my questions.

Author Response

There is no comment from this reviewer.